# How Many Trees Are Planted in African Cities? Expectations of and Challenges to Planning Considering Current Tree Planting Projects

**Emmanuel S. H. Lobe Ekamby [1,2,*] and Pierpaolo Mudu [3]**

1   Department of Geography, Bonn University, Meckenheimer Allee 166, 53115 Bonn, Germany
2   Institute for Environmental Risks and Human Security, United Nations University (UNU-EHS), Platz d. Vereinten Nationen 1, 53113 Bonn, Germany
3   Public Health, Environmental and Social Determinants of Health (PHE), World Health Organization (WHO), Avenue Appia 20, 1211 Geneva, Switzerland
*   Correspondence: s6emlobe@uni-bonn.de

**Abstract:** Africa is a particularly vulnerable continent to the effects of climate and environmental changes. Several tree planting projects were launched as one of the plausible solutions to climate action. This paper reviews the recent tree planting projects in African cities, initiated between 2009 to 2021, focusing on analyzing the expected benefits to the populations. Indeed, these projects have become a widespread instrument to promise planning solutions for many African countries given the expected improvements to mitigate air pollution, carbon sequestration, and the conditions of cities for the health and wellbeing of their citizens. The consequences of uncontrolled urbanization in Africa also stress the importance of better planning of green spaces. African cities should reconsider urban planning with a clear focus on the role of green infrastructures because of their extensive social benefits and supportive capacity for any significant sustainable development. While these projects seem to be a promising initiative and are expected to deliver beneficial ecosystem services to citizens, there are still some loopholes that this paper highlights.

**Keywords:** Africa; trees; nature-based solutions; urban planning; climate change; sustainable development

## 1. Introduction

Since the beginning of the 21st century, the world's population has experienced intense patterns of urbanization [1]. This demographic shift has positioned cities in a determinant role to guarantee the development and wellbeing of populations and their countries. Rapid urbanization trends pose severe threats to biodiversity and ecosystems on a local, national, and global level [2,3]. Africa is experiencing the fastest urbanization rate in the world, with around 43 percent of its population living in urban areas [4]. Twenty-one percent of Africa's urban population lives in coastal cities and 38% of agglomerations of 1 million inhabitants and above are coastal agglomerations [5]. It is well known that coastal cities are exposed to many climate risks, such as flooding, sea level rise, and groundwater salinization [6,7]. The continent already accommodates three megacities, Lagos, Cairo, and Kinshasa, and this number is expected to grow dramatically in the near future [4]. Furthermore, as cities continue expanding in Africa, the costs and benefits of urban green infrastructure are rarely considered and prioritized in urban planning and development processes [8].

Although there has been an increasing number of studies on green infrastructure, nature-based solutions, and ecosystem services in Africa [9,10], the information and knowledge available used to tackle the rapid urbanization is not sufficiently accompanied by policies and measures able to protect its ecosystems, fight climate change, and address the increasing pollution of its environment [11]. The evidence of green infrastructure

and ecosystem services benefits are broad [12], and particularly their effects on health and wellbeing are significant [13]. In fact, the multifaceted relationship between natural environments, green spaces, as well as human health and well-being has been the object of increasing scrutiny [13]. The presence of green spaces is associated with reduced mortality and positive effects on mental health [14–16]. While the importance of green spaces has been widely recognized, deforestation, loss of vegetation, and the absence of green spaces in urban planning have developed in parallel [17].

This paper aims to analyze the tree planting projects in African cities given the increasing pressure of expanding populations on limited resources and the subsequent environmental hazards. These are crucial challenges for cities to create healthy and sustainable living environments. It is worth monitoring and evaluating the impacts of tree planting, as part of a strong narrative on the importance of green infrastructure, within cities. In our analysis, this is done within the period 2009–2021. Reviewing tree planting projects allows for analyzing and evaluating their impacts within cities in relation to carbon sequestration, air pollution, water provision, socioeconomic, and health, among others. This paper seeks to answer the two main research questions: (1) What relevant information for urban planning can we obtain from policies of planting trees in African cities in the last decade? (2) What are the direct and indirect socio-environmental impacts expected by these tree planting projects?

## 2. Materials and Methods

A literature review and document analysis were carried out to collect the relevant information related to the topic. The data on policies of tree planting in Africa were collected from a wide range of sources. Two search engines were searched (Web of Science and Google Scholar) using terms such as "planting trees", "plant trees", "Africa", "English name of the country"; in French, using "plantation d'arbres", "planter des arbres", and the French name of the country; and in Portuguese using "plantação de árvores", "plantar árvores", and the Portuguese name of the country. General searches using the same keywords in Google also were carried out. Most of the data originate from official documents, national project reports, media reports, United Nations reports, scientific literature, and from international development agencies such as the German Agency for International Cooperation (Gesellschaft für Internationale Zusammenarbeit—GIZ) and non-governmental organizations like the Worldwide Fund (WWF), Arbor Day Foundation, OneTreePlanted, and Nature4Climate. Wherever possible, data were controlled by verifying information from multiple sources. Out of 54 countries in Africa, 181 tree planting projects, covering all African countries except for Guinea-Bissau, were identified, and 60 were analyzed for this paper. The criteria of inclusion of these projects were an urban focus and the availability of some details on the intervention, socioecological impacts, and reasons for planting, as well as on the responsible agencies, funds, and investors. On the contrary, criteria of exclusion of projects included the fact that they were not urban but regional or rural; a lack of information (for example unclear area of intervention); and although named as tree plantation interventions, they were not in reality tree planting projects. Tree planting projects from national arbor days were considered only when urban-centered. Continental massive green initiatives, such as the Great Green Wall Initiative in the Sahel and West Africa—one of the largest reforestation projects across 21 African countries [18], through the African Union—was only considered when involving urban areas, whereas most of the initiative is ongoing in rural areas.

## 3. Results

### 3.1. Summary of Projects Examined

The 60 projects that were selected for analysis have a specific urban focus. These projects are displayed in Table 1 and additional information can be found in the supplementary materials. These selected projects are part of the main analysis of our search, which identified 181 tree planting projects in Africa in the period between January 2009

and October 2021. The projects selected are often part of national or regional strategies of reforestation (e.g., in Cote d'Ivoire, Egypt, Ethiopia, Malawi, Mali, Nigeria, Rwanda, Seychelles, and Tunisia); in some cases, they part of a single urban tree planting project without a regional or national connotation; and in a few cases, they were part of international initiatives (e.g., in Mauritania or Niger). From our perspective, the national, and multi-scalar nature of many projects is associated with the strong influence of government leaders and authorities who seek to use tree planting as a propaganda tool for political advantages and national power consolidation. Power relations and propaganda patterns are aspects that deserve more detailed future analysis.

The location of projects has also to be understood in the context of the wide variation in rainfall and water availability across the continent, given the fact that many areas lack adequate rainfall to support large-scale tree-planting, including much of southern Africa and the Sahelian region [19,20]. In any case, the projects identified and analyzed indicate an overall concentration in areas with lower levels of total annual rainfall. (Tree planting projects are distributed within African cities, although the countries in Central Africa host fewer projects than the rest of the continent. These tree planting projects appear on a map in the supplementary materials.

Unfortunately, we cannot compare our results with similar studies. Nevertheless, in comparison to the analysis carried by Jindal et al. [19] of 1992–2008 projects, we observed that projects have increased in number and geographical coverage, and also the potential benefits considered are wider that in the previous analyses. Urban tree planting projects have been increasingly encouraged in countries such as South Africa, Nigeria, Egypt, Tanzania, and Morocco, which have stronger economies and large cities than other African countries. South Africa is an example where many urban tree projects have comprehensive publications and detailed information is available for its major cities, such as Cape Town, Durban, and Johannesburg [21–23]. It is not easy to categorize if projects are essentially political announcements, commercial, or benefit local communities, but the declared aim of most of the projects is to fight climate change through carbon sequestration. Besides reducing carbon emissions, there are other objectives that are interrelated to the main aim, such as biodiversity conservation (e.g., Burkina Faso, South Africa, Benin, and Nigeria), combatting desertification (see the example of Ouarzazate in Morocco or in cities in the Republic of Congo, Egypt, Ethiopia, Mali, Nigeria, and Tunisia), and in some cases, multi-sectoral impacts are expected in the environmental, social, economic, and health domains (see Supplementary Materials).

Many tree planting projects did not openly mention their attachment to the Kyoto Protocol and Paris Agreement, but their activities portrayed their compliance. Some projects were initiated and executed to fulfill the 2030 Agenda (UN Sustainable Development Goals) and the Great Green Wall Initiative.

On one hand, many tree planting projects were launched in capitals, such as Accra, Addis Ababa, Bamako, Freetown, Kigali, Lomé, Nairobi, and others, whereas on the other hand, some projects were done in other cities, for example, Ismailia, Marrakesh, and Tshwane. Besides, there were numerous tree planting projects launched in large cities, for example, Lagos (not political capitals), as compared to medium sized cities such as Cape Town and Johannesburg (Table 1 and Supplementary Materials).

**Table 1.** Overview of projects to plant trees in African cities (ordered by country).

| | Country (City) | The Number of Trees to Be Planted or Already Planted | Duration of Plantation |
|---|---|---|---|
| 1. | Algiers (Algeria) | 25 million trees planted at the National Arbor Day announcement, and 17 million were expected to be planted by March 2020, in line with the national tree program of 43 million trees.<br>Number of trees not mentioned in the master plan that aims at transforming the city into a sustainable city with a garden city implanted within it.<br>Number of trees not mentioned, but there is a project aimed at transforming the Oued Smar landfill (30 ha) into an urban ecological park. This project falls into to the major green plan of Algiers of 2035. | Algiers 1 Tree project:2019–2021<br>Algiers 2 Tree project: 2013–2030<br>Algiers 3 Tree project: 2009–2018 |
| 2. | Baraki (Algeria) | More than 2000 trees were planted after the announcement and 1 million trees are expected to be planted in one year. | 2020 |
| 3. | Luanda (Angola) | 1500 trees were planted in the city.<br>Number of trees not mentioned, but in Rangel (neighborhood of Luanda) there is a trees planting project announced during the national trees day. | Luanda 1 Tree project: 2018<br>Luanda 2 Tree project: 2019 |
| 4. | Andulo (Angola) | 300 trees were planted. | 2020 |
| 5. | Huambo (Angola) | 1000 trees were planted, and 2000 trees were expected to be planted. | 2020 |
| 6. | Allada (Benin) | 2500 trees were planted at the announcement. | 2019–2020 |
| 7. | Parakou (Benin) | 2100 trees were planted at the announcement by the Beninese government and municipal authority. | 2019–2020 |
| 8. | Savé (Benin) | 1250 trees were planted at the announcement. | 2019–2020 |
| 9. | Gaborone (Botswana) | 300 trees were planted. | 2013 |
| 10. | Bobo-Dioulasso (Burkina Faso) | The number of trees not mentioned, but 6.9 hectare of green spaces were planted. | 2012–12014 |
| 11. | Ouagadougou (Burkina Faso) | 1000 trees were planted, and 80,000 trees are expected to be planted. | 2019–ongoing |
| 12. | Pissa (Central African Republic) | 12,000 trees were plannted and 300,000 trees were to be planted afterwards. | 2019–2020 |
| 13. | Brazzaville (Congo Republic) | 160,000 trees planted. | 2011–2021 |
| 14. | Abidjan (Cote d'Ivoire) | 500 trees planted in 2019 along the road of the airport Port-Bouët within 3 km long, 400,000 trees in Abidjan and the 2.1 milllion planted nationwide. | 2019–2030 |
| 15. | Cairo (Egypt) | 12,000 trees planted, and 1 million trees to be planted by 2019, as part of a national project called the Egypt's 1 Million Trees.<br>350 trees were planted, and 14,000 shrubs were also added for a vertical forest.<br>The number of trees not available but trees are a fundamental part of construction of the capital park located in the under construction Egypt's New Administrative Capital close to Cairo. | Cairo 1 Tree project: 2019.<br>Cairo 2 Tree project: 2020–2022.<br>Cairo 3 Tree project: 2016–2030. |
| 16. | Ismailia (Egypt) | 240 hectares of land have been reclaimed for trees planting. 500,000 hectares of desert land could be reclaim for afforestation. | 2012–ongoing. |

**Table 1.** *Cont.*

|  | Country (City) | The Number of Trees to Be Planted or Already Planted | Duration of Plantation |
|---|---|---|---|
| 17. | Massawa (Eritrea) | Number of trees not mentioned. Tree planting activities are being conducted in the port city of Massawa in connection with 30th anniversary of the commemoration of Operation Fenkel. | 2020 |
| 18. | Addis Ababa (Ethiopia) | More than 350 million trees were planted nationwide in a day starting from the capital as part of the Green Legacy initiative. Number of trees not mentioned, but there is also the "Beautifying Sheger" project to creating a 56 km green spaces along the river. The city is expected 5 million trees are to be planted within the 20 billion trees nation plan. | Addis Ababa 1 Tree project: May-October 2019. Addis Ababa 2 Tree project: 2019-ongoing. Addis Ababa 3 Tree project: 2019–2024. |
| 19. | Accra (Ghana) | 100,000 trees planted in Accra which is part of 10 million trees of the national tree program. | 2019 |
| 20. | Nairobi (Kenya) | 1.8 billion trees are expected to be planted in Nairobi and nationwide. About 10,000 trees are expected to be planted. 1500 trees were to be planted in the Ngong Forest (25 km from Nairobi) after the announcement. | Nairobi 1 Tree project: 2018–2022. Nairobi 2 Tree project: 2020–2023. Nairobi 3 Tree project: 2021. |
| 21. | Daadab (Kenya) | 52,000 indigenous tree seedlings, in the refugee camp 'IFO2' in Daadab. | 2018 |
| 22. | Maseru (Lesotho) | Around 200–300 trees were expected to be planted in the capital and nationwide. | 2020 |
| 23. | Monrovia (Liberia) | 10,000 trees were planted. | 2017–2019 |
| 24. | Antananarivo (Madagascar) | 1 million trees were planted in the span of a few hours. 60 million trees are expected to be planted. | 2020 |
| 25. | Lilongwe (Malawi) | 62 million trees are already planted and more than 60 million trees are expected to be planted. 1 million trees were planted at the Dzalanyama Forest Tree Project. | Lilongwe 1 Tree project: 2020–2021. Lilongwe 2 Tree project: 2017–2018. |
| 26. | Bamako (Mali) | 18,000 trees are to be planted in the city. This is included in the national tree program of 22 million expected to be planted by 2023 under the initiative "Un Malien, un arbre". 150 trees were planted in the Garden of Eden in Bamako. | Bamako 1 Tree project: 2019–2023. Bamako 2 Tree project: 2019. |
| 27. | Nouakchott (Mauritania) | As part of the Great Green Wall Initiative, 2 million trees are expected to be planted, and 200,000 trees were already planted in 2010. | 2010–2014. |
| 28. | Port Louis (Mauritius) | 1 million trees to be planted, which means 50,000 trees per year. | 2018–2030 |
| 29. | Marrakesh (Morocco) | 3 million trees were planted. 800,000 trees were expected to be planted before the end of the year. | 2017 |
| 30 | Jbilat (Morrocco) | 1 million trees to be planted. | 2016 |
| 31. | Ouarzazate (Morroco) | About 635 hectares of trees were planted to act as a protective buffer between the city and the desert. | 2012–2017 |

**Table 1.** *Cont.*

| | Country (City) | The Number of Trees to Be Planted or Already Planted | Duration of Plantation |
|---|---|---|---|
| 32. | Katembe & Madjuva (Mozambique) | 55,0000 trees have already been planted and 750,000 trees are to be planted. | 2018–2020 |
| 33. | Windhoek (Namibia) | 500 trees were planted. | 2019 |
| 34. | Niamey (Niger) | 20,000 trees were planted in support of the Great Green Wall Initiative. 3 million trees were planted to address heat waves and pollution. | Niamey 1 Tree project: 2018–2020. Niamey 2 Tree project: 2019–2020. |
| 35. | Abuja (Nigeria) | 30 million trees to be planted as a part of the Presidential program starting from the capital and nationwide. | 2020 |
| 36. | Sokoto (Nigeria) | 1 million trees were planted in (2016). | 2016 |
| 37. | Lagos (Nigeria) | The number of trees planted is not mentioned, but there is a 5-year tree planting green project, and specific projects also to transform the Olusosun dumpsite (100 ha) into a green space. | 2014–2024 |
| 38. | Kigali (Rwanda) | The number of trees mentioned, but 43,589 hectares of trees were expected starting from Kigali to other parts of the country. | 2018–2019 |
| 39. | Dakar (Senegal) | 1300 trees will be planted in the public park, located near the Ouakam Divinity Mosque in Dakar (2020–2035). An urban forest park covering 10 hectares was planted on the site of the former Dakar airport (2020–2035). | Dakar 1 Tree project: 2020–2035. Dakar 2 Tree project: 2020–2035. |
| 40. | Victoria (Seychelles) | 4000 trees were expected to be planted in this city and other parts of the country. | 2018 |
| 41. | Freetown (Sierra Leone) | Over 12,000 trees are expected to be planted. | 2019–2023 |
| 42. | Mogadishu (Somalia) | 4000 trees were planted in the capital city during Arbor day in (2013). | 2013 |
| 43. | Johannesburg (South Africa) | 90,000 trees were planted and 200,000 trees are expected to be planted (in particular in Soweto) in support of the 2010 World Cup Green Program. | 2009–2010 |
| 44. | Riverside View (South Africa) | 2000 trees were expected to be planted. | 2019 |
| 45. | Riverlea (South Africa) | 200 trees were planted and 5000 trees to be planted. | 2020 |
| 46. | Cape Town (South Africa) | 210 mature trees were planted within 2 months before the World Cup started (2010). 15,523 trees were planted in 358 beneficiary sites (2010–2017). | Cape Town 1 Tree project: 2010. Cape Town 2 Tree project: 2010–2017. |
| 47. | Tshwane (South Africa) | 115,200 trees were planted. | 2002–2032 |
| 48. | Durban (South Africa) | 10,000 trees were planted for the Buffelsdraai Tree Planting project. | 2019 |
| 49. | uMhlathuze city (South Africa) | 10 million trees are expected to be planted in the next five years in South Africa. This trees planting project was launched during the recent tree day in September 2021. | 2021–2026 |
| 50. | Al Jabalain (Sudan) | 1 million trees were planted in a refugee camp by UNHCR. | 2017–2020 |

**Table 1.** *Cont.*

| | Country (City) | The Number of Trees to Be Planted or Already Planted | Duration of Plantation |
|---|---|---|---|
| 51. | White Nile State (Sudan) | 1 million trees were planted. | 2017–2021 |
| 52. | Khartoum (Sudan) | The number of trees not mentioned for one of the few remaining urban forest reserve in Sudan. | 2018 |
| 53. | Dar es Salaam, Dodoma, Arusha and others (Tanzania) | 4000 trees planted and 50 million trees to be expected. | 2016–2020 |
| 54. | Dodoma (Tanzania) | Number of trees not mentioned (but the type of trees planted specified) for a project aimed at transforming the semi-arid Dodoma into a green city.<br>4000 trees are expected to be planted to support the government efforts in making Dodoma city green. | Dodoma 1 Tree project: 2020.<br>Dodoma 2 Tree project: 2021 |
| 55. | Lomé (Togo) | 50,0000 trees planted during the national arbor day. | 2020 |
| 56. | El Agba (Tunisia) | Number of trees not mentioned, but 3 ha of trees planted. | 2018–2019 |
| 57. | Kasserine and Jendouba (Tunisia) | 2 million trees will be planted in the areas most affected by fires in the summer 2021. This is a national reforestation program which focus mainly on these two cities. | 2021–2022 |
| 58. | Chinsali (Zambia) | 1 billion trees were expected to be planted in 2019. 2 billion trees planned by the governement in three years timeframe. | 2019–2021 |
| 59. | Harare (Zimbabwe) | Number of trees not mentioned, but the project is greening the "Sunshine City".<br>15 million trees expected to be planted on the 1st December (national tree day). | Harare 1 Tree project: 2017<br>Harare 2 Tree project: 2018 |
| 60. | Addis Ababa-Djibouti (Ethiopia and Djibouti) | Trees are not mentioned, but there is a tree-planting campaign themed "Green Addis Ababa-Djibouti, Green Ethiopia" as a way to respond to the national Tree-planting campaign along the railway of these countries. | 2020–ongoing. |

### 3.2. The Announcement and Implementation of Projects

Even though it is difficult to dissect between those tree planting projects' announcements that are for political propaganda or political will, there is an indication that tree planting projects play an important role for many African governments and national and geopolitical agendas play a significant role to announce tree planting projects. The announcement of tree planting projects is often carried out by African Presidents or national governments. An "Arbor Day" to encourage trees planting is celebrated in many countries in different days and with various names. Niger has celebrated its Arbor Day as part of its Independence Day since 1975, but the adoption of this day to mark the calendar has been more recent in most African countries. Some 200,000 new trees were planted across Burkina Faso in connection with the first edition of its National Tree Day (NTD) scheduled on 3 August 2019, and presided by the President of Burkina Faso, Kaboré, during the launching ceremony. In Freetown, over 12,000 trees were expected to be planted by the end of 2019, with the aim to improve the vegetation until 2023, in accordance with the World's Environment Day under the theme "Air Pollution". In South Africa, every first week of September (1st–7th) marks the National Arbor Day, which has been tradition since 1983 [24], and projects were also implemented for the 2010 World Cup.

In May 2019, the Ethiopian Prime Minister announced via a Tweet the planting of 4 billion trees by the end of the rainy season of the same year, and on 29 July 2019, he launched a tree planting project in Addis Ababa, where over 350 million trees were planted nationwide within 12 h to counter the effects of deforestation and climate change [25]. The Nigerian President Buhari announced the planting of 25 million trees in his country during the UN Climate Action Summit in September 2019. In an article to present the project, the Nigerian President recalled that this contributes to the Great Green Wall Initiative and that "We must use nature's basic material (all too often overlooked) for solutions to one of the continent's greatest challenges" [26], considering the importance of trees to the Nigerian society and the Sahel for their multiple benefits to citizens who are suffering from climate change effects and desertification. Concluding: "The time for solely cutting emissions has ended. Sequestering the carbon already in the atmosphere must move to the front of efforts and the best technology we have for this is trees. Nature's processes can heal what man has wrought. And in acting as a barrier to an encroaching desert, it offers a gateway to collective peace and prosperity across the Sahel" [26]. There is in other cases the narrative of transforming current cities into "green cities" [4]. This is the case for Accra, Algiers, Bamako, Dodoma, and Harare. This emphasis on "green cities" is also present at the international level; the FAO, for example, has a green cities initiative with a "Regional Action Programme for Africa" signed by Praia (Cabo Verde), Kisumu and Nairobi (Kenya), Antananarivo (Madagascar), Quelimane (Mozambique), and Kigali (Rwanda) [27].

### 3.3. The Organization of Projects

Current tree planting projects in Africa pay attention to being compliant with the international climate agenda and are involved in collaboration with key players such as investors, local organizations, and governments. The roles of the actors can be unbalanced when executing tree planting projects, as investors are greatly involved in both sponsoring and executing these projects compared to governments. Examples can be seen in Ismailia (Egypt), where foreign investors (German Academic Exchange Service—DAAD) provide both financial and technical support to plant trees (see Supplementary Materials). International donors are sponsors in 20 out of 60 cities and are active as responsible agencies in most of these projects. NGOs are also other players active both at the local and international scale (13 out of 60 cities); for example, in Niamey (Niger), tree planting project related to the Great Green Wall Initiative involves the Aid Tree NGO. The reference to the international agenda seems to have obscured the past situation that had several projects, sponsored by multinational timber companies, meant for commercial benefits rather than environmental purposes [19].

*3.4. The Expected Outcomes of Projects*

The expected outcomes of tree planting projects can be quantified in terms of the number of trees and/or in terms of area size; this information is often missing. The potential positive effects on the health and well-being of trees are often unmentioned. A summary of the expected outcomes of some of the studied tree planting projects can offer other points of debate (see Table 1 and Supplementary Materials). Some projects are targeting climate change objectives, with long-term perspectives (40 out of 60 cities). Tree species are mentioned in a few projects; for example, cashew nuts, mangoes, sisal, and grapes for Dodoma, and the best solutions and selection of indigenous trees were discussed in projects in Nairobi. In the case of Marrakesh, about 3 million Argan trees were planted along the roads to fight climate change and reduce air pollution. The trees' planting project is both funded and implemented by the government through the Moroccan Highways authority supported by the Department of Agriculture. Argan trees are an indigenous species that is expected to absorb the carbon emissions from emissions of cars and trucks driving along the roads of the city. In Ismailia (Egypt), German donors decided to regenerate dried-up forests in this city by reclaiming 240 hectares for tree planting.

Some urban projects had specific sites for tree planting. For example, in 2017, 10,000 coconut and palm trees have been planted along the Tubman Boulevard of Monrovia. The main reasons are to beautify the city, for air purification, and to fight heat waves. In some cases, the integration of the tree planting projects within the master plan of cities is also indicated, but details are not easily accessible; for example, for Algiers or Egypt's new administrative capitals. In the case of Egypt, planting trees policies are controversial, and the official data issued by the Egypt Central Agency for Public Mobilization and Statistics (CAPMAS) estimated a very poor availability of green spaces for the population, and there is the ongoing destruction of trees and removal of green spaces, especially in Cairo, to support motorized traffic and new roads to the new administrative capital [28].

The Lagos State Parks and Gardens Agency (LASPARK) developed a five-year master plan (2015–2020) for tree planting in the state and Lagos city in particular. Although the number of trees was not mentioned, the authorities announced that this tree planting project falls into the urban plan of Lagos city and carbon sequestration will be achieved effectively [29]. In Harare, the city council embarked on greening the city by planting an unknown number of trees in 2017, and this initiative is aligned with the city's master plan. Harare, "the Sunshine City", wants to reclaim its lost glory following decades of economic decline in the country [30]. In 2018, 15 million trees were expected to be planted in Harare as part of National Tree Day (1 December every year). This tree planting falls within the plan of the city though it is a national initiative [31].

*3.5. The Use of Funds*

Investments in tree planting projects cover a big range. Small-scale projects are carried out with investments below EUR 100,000. The authorities of the city of Bobo-Dioulasso (Burkina Faso) hosted 6.9 hectares of green space invested by UN-Habitat, which cost EUR 20,000. The Urban Forest of El Agba (Tunisia) project (2018–2019) consisted of planting 3 hectares of forest in an area that is constantly degrading. The cost of this project is EUR 60,000, and it was launched by the Forest Department of Tunis.

Medium-scale projects can range from 100,000 to 1 million EUR. The Lilongwe Tree Project (Malawi), to restore the degraded Dzalanyama Forest reserve with 1 million trees, costed USD 100,000, which came from both the city and national authorities but mostly the Japan International Cooperation Agency (JICA). The Ismailia project (Egypt) of 240 hectares of trees planted costed EUR 260,000 (approximately USD 285,600), which came from the German Academic Exchange (DAAD).

In other cases, investments are over EUR 1 million. For example, in Abidjan (Cote d'Ivoire), the planting of 450,000 trees costed EUR 939 million (approximately USD 1,031,646,000), coming from the Ivorian government. In Tshwane (South Africa), the planting of 115,200 trees costed USD 3 million. The authorities of the city of Yie (Re-

public of Congo) planted 160,000 trees at a cost of USD 2.5 billion, mostly from donor countries. The Kigali Tree project (Rwanda) is part of the national tree campaign of 2018/19, which entailed planting millions of trees, with a total of 43,589 hectares of land set to be covered with trees over the next six months. The cost of this project was RWF 4.4 billion (approximately USD 4,704,752.80). The Grand Niamey (Niger) urban development, which consists of sanitizing and greening for the 33rd African Union Summit 2019, cost CFA 31 billion (approximately EUR 47 million). On average, a rough estimate of planting one single tree per project costs between 10 and 100 USD.

Innovative solutions are also presented but difficult to appraise; for example, the first example of a vertical forest in Cairo or considerable investments in refugee camps. In refugee camps run by the UNHCR, planting of trees has been a recent activity; for example, in the Dadaab refugee camps that host more than two hundred thousand inhabitants (Kenya), or in Al Jabalain (Sudan), which hosts fifty-nine thousand people. In Niamey, a project is fighting desertification by using migrants under the supervision of the International Organization for Migration (IOM).

## 4. Discussion of Results

The discussion of results is organized into three sections.

### 4.1. Multi-Scalar Planning and Urban Governance

It is difficult to separate tree projects from more general problems of sustainable development, but the focus on tree plantations restrict the attention on some specific policies that have been announced and implemented in Africa and its cities and this allows to discuss urban planning approaches applied in different cities. There is a multi-scalar nature of many projects that are implemented to fulfill national, regional, and local needs within global frameworks. The global scale is assumed when tree planting is an act to landmark international agreements. Projects are designed to construct their scale that is not to be understood as a fixed spatial configuration, an established platform, a given container for the social activities of plantation of trees, but as a process that is negotiated within multiple layers [32].

Urban areas are strategic arenas for environmental policies and urban governance of green spaces is carried out through discourses and interventions that reflect the conditions of specific contexts. Nevertheless, there is a common thread that embeds different projects. Urban green infrastructure and its ecosystem services are often conceptualized in terms of a predominantly Western perspective of cities [33]. In fact, since the end of colonialism, policies on "artificial and natural" regeneration have been implemented in Africa [34,35]. Desertification and climate change have dictated a political agenda that has produced hundreds of different uncoordinated projects [19]. Additionally, the whole issue of participation in policies and their implementation is rarely addressed [36,37].

All the projects examined and the urban governance that is implied in them refer us to consider and expand the concept of green sovereignty [37]. Green sovereignty is the willingness and eagerness for a country to regain its vegetation cover and natural ecosystem lost due to the pressure of economic growth and socio-economic demands of the population. Green sovereignty is exercised in several ways, currently mainly through the planting of trees on a large scale. Green sovereignty urges the need of acknowledging and implementing environmental protection in the national development plans of states [37]. However, the risk is to enter in a green propaganda that gives short-term announcements without long-term sustainable directions. In addition, it should be noted that even if the State is sovereign over its given territory, usually the scale of implementation of green projects is done at the local level as well. To illustrate this, the announcements of million-tree plantations are quite abstract and difficult to visualize whereas the work done in rural communities or at the urban level has a level of visibility and high public scrutiny that makes it crucial to envisage different approaches to planning. In this sense, green sovereignty plays a role that is not just the one of the states as owner/main ruler of the

territory but the one of citizens that collectively preclude the destruction of their ecosystems and prevent the different institutions that have interests in specific territories to "have a positive obligation to act as environmental caretaker" [37].

*4.2. Outcomes and Impacts*

The projects analyzed gave a vast range of contradictory outcomes. On one side many governments announced the green economy as their new target, capital cities are the starting point for reforestation of countries, and there is the idea of diversifying the economy and stopping the exploitation of natural resources; on another side, projects lack fundamental details (e.g., types of trees and tracking mechanisms), green spaces are a major element for political propaganda, and urban governance is revealed as an extreme top-to-bottom planning approach. Often, the chronic lack of funds and poor management of resources is embedded in greening city projects, to attract foreign direct investments.

Impacts on the ecosystem are a central issue to be considered, as it is concerned with evaluating the post-planting period in the respective African countries [38]. Many tree planting projects in Africa mentioned the expected socioeconomic impacts (job creation, increase people's income, food security, environmental education, and boosting the economy) while very little information on carbon sequestration, health, air, and water impacts were mentioned though considered. For example, in Senegal, with the "Plante ton arbre" regional project launched in 2009, an assessment of the expected impacts (social, health, air, water, and carbon sequestration) was done in terms of generating income activities for youths; also, ecosystem-based activities, such as a sustainable form of fishing, were encouraged, and building the capacity of the local people on biodiversity was achieved [39]. However, this is not usual for many tree planting projects in Africa, which clearly mention the expected impacts in the initial stage but later give little information if these impacts were achieved or not.

In addition, even though there is little information on impacts such as carbon sequestration, this paper can present some carbon sequestration data from a few of these tree planting projects (carbon sequestration data are found in the Supplementary Materials). For instance, in the case of Ismailia (Egypt), over 25 million tons of carbon dioxide offset are expected from this afforestation project. Another example is a tree planting project in Abuja (Nigeria), whereby around 74 million tons of carbon dioxide per annum are expected until 2030. A tree planting project in the city of Tshwane (South Africa) resulted in about 200,492 tons of carbon dioxide sequestration since its completion and more carbon sequestration are expected by 2032. Thus, these carbon sequestration data collected from these countries (Egypt, South Africa, and Nigeria) could portray the increasing awareness of governments of these countries to achieve sustainable development by encouraging tree planting projects within cities and measuring their socioecological impacts in the future.

*4.3. Challenges of Trees Projects*

The studied tree planting projects executed in the last decade in Africa depict the increasing willingness of governments and cities to join the fight against climate change, but several challenges are left unresolved; for example, the challenges faced when executing these projects include insufficient funds, inadequate tracking policies, unknown carbon sequestration estimated, and expected socioeconomic impacts.

4.3.1. Challenges Related to Data Availability

A first challenge to consider is a dire shortage of public data about tree planting projects in Africa. This shortage of data can be translated into an insufficient number of scientific publications, and thus academic literature that discusses intensively and entirely the urban green infrastructure in Africa [33]. The few available documents are mostly penned by researchers of the Global North, who coined urban concepts and eco-urban planning in Africa from a Western perspective, sometimes ignoring African realities [33]. Second, most projects studied had insufficient information on health considerations, air

pollution, water impacts, and even funds invested. This insufficiency in information made the task of evaluation very challenging, also considering the evidence accumulated so far [40]. Third, many African governments do not publish their green space projects on their official websites, and even their annual reports have less data on tree planting projects.

Moreover, tracking policies and measures of tree planting projects in Africa announced by many African Presidents during international summits are missing. The absence of monitoring and tracking systems do not allow to tackle the usual problems of the urban environment, such as water scarcity, weed competition, and man damage, if there is a lack of maintenance of trees for early detection of stress and diseases [41,42].

There is a significant research gap in the analysis of the allotment and use of funds for tree planting projects. There is the need to investigate in detail the role of institutional agreements and frameworks in which environmental policy and governance are currently embedded in orienting and shaping the value of some projects instead of others. This is relevant because of the wider "economic and sociopolitical processes that have governed the expansion of pricing into previously non-marketed areas of the environment" [43]. There are questions that further research should answer; e.g., How much ecosystem protection policies can help fight poverty and how can tree-based economies be sustained?

Considering all the mentioned challenges, the main limitation of our study is related to the fact that our analysis relied on publicly available data. Although projects might exist and are ongoing, the information could not be available for some places and can be very partial.

### 4.3.2. Suggested Critical Points for Sustainable Urban Green Infrastructure Planning

It is important to summarize some critical points to the urban problems in African cities to encourage green urban infrastructure policies to be implemented effectively.

The situation of deprivation and dispossession for many countries makes them subject to the geopolitical agenda that constitutes heavy pressure and constrain generated by the most-polluting countries on the planet. As Africa is urbanizing rapidly, governments and mayors should seize this urban opportunity by transforming African cities into sustainable cities. This transformation can only be possible if green infrastructures are considered a fundamental asset. This greening of urban areas can directly create positive socioeconomic outcomes and ecosystem services [19,33]. The health benefits of neighborhood greenness suggest greening strategies could be considered as part of broader public health interventions for non-communicable diseases [44].

Planting trees and green projects in African urban areas should not only be initiated from political initiatives but communities should be encouraged to cater to their environment given the current ecological and urban challenges the continent is experiencing. Tree projects should adopt a bottom-up approach, whereby the population is consulted from the initial to the maturity of these projects. A community-based approach also should be highly encouraged when executing these projects. This will create more awareness in the minds of the population about sustainable urban development and the expected challenges these populations might face. A bottom-up approach will lead to an effective outcome, wherefore the target population will be the main beneficiaries and not public or private companies, as is usual [45]. Governments should allocate sufficient funds for these projects within the national annual budget and even create laws to enforce environmental protection within cities and nationwide.

To facilitate access to data on urban green infrastructure and ecosystem services in Africa, it will be interesting for future green urban projects to have annual reports where important data on indicators such as health consideration, carbon sequestration, water impacts, tracking policies, socioeconomic impacts, and others are clearly mentioned and used for public scrutiny and scientific research.

These political considerations cannot produce any beneficial impact if the projects do not consider that the urban environment significantly differs from the non-urban one: temperature is generally higher and soil compaction, waterlogging, water stress, pollution,

altered light conditions, and restricted rooting space are in many cases the environment in which trees will be planted [41,46]. Like what is done in city planning in North America or Europe, there is the need to carefully consider the selection of tree species, considering also local indigenous tree availability and the potential influence on insects [47].

The terminology and definitions used to introduce the tree planting projects are also relevant. In the published literature of the academic analysis of tree planting projects, terms such as urban green infrastructure, urban greening, urban agriculture, nature-based solution, and urban forestry are used interchangeably [2,48,49]. The use, or absence, of precise definitions has many implications in terms of assumptions and conceptualizations of the policies that constitute a relevant area of research.

Lastly, African governments, through their leaders, should not just stop at announcing massive tree initiatives before the international community, and then subsequently not properly realizing them and not stopping the deforestation patterns. Tree planting projects should not be used as political propaganda but instead included in the National Development Plans (NDP) of African countries, to effectively tackle climate change.

To summarize, from this review and analysis of projects, some indications for urban planning emerged clearly on planting trees in cities:

- The geographical context matters and cannot be ignored: indigenous species of trees, latitude, altitude, lack of precipitation, and proximity to the coast all influence the organization and potential positive outcomes of projects.
- The health and social impacts of projects should be taken into consideration to discuss all the implications of such projects.
- Policy monitoring and tracking, which is the necessary to have a systematic multidisciplinary approach to identifying, monitoring, and evaluating the implementation and progress of a policy in time and space, should always be included.

These indications are likely to be similar to other steps that need to be taken into account when planning to address environmental policies, but there are very specific elements related to urban forestry and urban green spaces that are quite unique.

## 5. Conclusions

In the last decade (2009–2021), Africa has seen tree planning projects at different scales and particularly within cities. This article analyzed 60 tree planting projects in Africa over the last decade (2009–2021). From an urban perspective, measures, policies, outcomes, and activities were analyzed and evaluated to verify how much these projects provide information on their effectiveness against climate change. Governments are aware that climate change is a threat to their development, and therefore they are aligning their national development plans to international agendas such as the Paris Agreement, the Sustainable Development Goals Agenda, and the Green Wall Initiative. Nevertheless, in encouraging urban green projects in their countries, they should be aware that cities are not just platforms to implement interventions, but they are complex ecosystem with governance issues. African cities are particularly vulnerable in terms of sustainability and health. Maintaining and improving ecosystems are crucial for achieving urban sustainability [50]. Green and healthy cities are a necessity for an urbanizing Africa. However, governments should be aware that tree plantations cannot be done in a haphazard manner, as it can limit other land-use patterns and activities relevant to sustainable livelihoods [51].

There is widespread knowledge and intuition that trees and green spaces provide health benefits, but it is difficult to argue at the technical level with precise arguments. Still, an anthropocentric vision, whereby human needs are prioritized over those of other species and ecosystems, is prevalent. If proper tracking measures are not implemented within tree projects, this might imply negative effects, such as soil acidification and decreased soil fertility in the long term, given that afforestation is highly depending on soil nutrients [51]. Our review suggests that only a huge, unprecedented effort driven by African countries and cities can stop the current trends in the management and destruction of nature.

**Supplementary Materials:** The following supporting information can be downloaded at: https://www.mdpi.com/article/10.3390/urbansci6030059/s1, Table S1: Details of trees planting projects in African countries; Figure S1: Map of Africa displaying the different tree planting projects.

**Author Contributions:** Conceptualization and methodology, P.M.; data collection, E.S.H.L.E.; writing—original draft preparation, E.S.H.L.E. and P.M.; visualization, E.S.H.L.E.; supervision, P.M.; writing—reviewing and editing, E.S.H.L.E. and P.M. All authors have read and agreed to the published version of the manuscript.

**Funding:** The work was partially funded by the Government of Norway through its financial contribution to advance the WHO's work on air pollution and health.

**Institutional Review Board Statement:** Not applicable.

**Informed Consent Statement:** Not applicable.

**Data Availability Statement:** Not applicable.

**Acknowledgments:** This paper reflects the authors' views. The authors alone are responsible for the views expressed in this publication, and they do not necessarily represent the views, decisions, or policies of the WHO. We thank Simone Borelli and Beatriz Kauark Fontes (FAO) for useful comments on the dataset of projects.

**Conflicts of Interest:** The authors declare no conflict of interest. The funders had no role in the design of the study; in the collection, analyses, or interpretation of data; in the writing of the manuscript; or in the decision to publish the results.

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
