# Peer review of "How Many Trees Are Planted in African Cities? Expectations of and Challenges to Planning Considering Current Tree Planting Projects"

_urbansci, doi:10.3390/urbansci6030059_

Round 1

Reviewer 1 Report

Dear authors,

Thank you for an interesting and meaningful topic. Overall, the manuscript is well prepared and written. Therefore, I suggest correcting some minor points as follows

In reference, the name of the journal is not identical. Therefore, the authors are suggested to thoroughly check and make them in journal format. (e.g., ref. 7; ref. 45; ref. 48…)

In SI, there are a number of typos (e.g, CO2...). Please check and correct these typos.

Reviewer 2 Report

Line 32: Write 21% in words as it starts the sentence

Line 48: "environments, green spaces, and human health and well-being" replace the first and with as well as

Line 69: Dedicated web engines such as Web of Science and Google scholar. Correct to : "Two search engines were used (Web of Science and Google scholar)" as there is no clear indication that other search engines were used. Also remove the s from Google scholars  

Line 76: GIZ: What is this abbreviation? 

Under materials and methods or results section: A distribution map of the countries that formed part of the analysis would be more informative 

Under results_3.1 Summary of projects examined: "The 60 projects that were selected for analysis have a specific urban focus (Table1 and 99 Supplemental material). These selected projects are part of the main analysis of our search 100 which identified 181 planting trees projects in Africa in the period between January 2009 101 and October 2021" Statistical analysis would be more informative to determine the significance of the 181 tree planting projects across 54 countries 

Line 106 to 107: "The national, and multi-scalar nature of many projects is associated with the strong influence of government leaders and authorities who seek to use tree planting as a tool for political advantages and national power consolidation" Is there any supporting evidence for this statement? If it is based on personal observation then it should be stated as such. 

Line 109-111: "The location of projects has also to be understood in the context of the wide variation in rainfall and water availability across all the continent, given the fact that many areas lack adequate rainfall to support large-scale tree-planting, including much of Southern Africa and the Sahelian region" Is there a  correlation between rainfall patterns and the tree planting projects conducted by different countries? 

Line 113: "although Central Africa hosts fewer projects than the rest of the continent" s there a specific reason behind this country planting fewer trees as compared to other countries? 

Line 132:  "Kigali, Lomé, Nairobi, and others, and on the other" Replace the second and with "whilst or whereas" 

Table 1 needs to be revised (separate columns for number of trees to be planted or already planted as well as for the proposed duration of plantation)

Line 191: NGOs. Put "The" before NGOs or write NGO in full 

Line 289: (Jindal, Swallow, and Kerr, 2008). Repalce with [19]. 

Reviewer 3 Report

This paper examines recent tree planting projects in African cities initiated between 2009 and 2021, with a focus on analyzing the expected population benefits. This is a well-written paper. However, I have some minor comments on this paper.

1. The findings of this study should be compared with those of previous studies.

2. The limitations of this study are negated. The authors should mention the limitations that they faced during their study.

3. Any quantitative results would improve the quality of the paper. 
